# A comparison of microbial community composition in two alpine springs in southern Nevada

Shawna Hunnicutt[1,2], El Hachemi Bouali[1]*, Bryan Sigel[1]

**1** Nevada State University, Henderson, Nevada, United States of America, **2** US Forest Service, Fredonia, Arizona, United States of America

☉ These authors contributed equally to this work.
* elhachemi.bouali@nevadastate.edu

## Abstract

Springs are intersections between the geosphere and hydrosphere where localized ecosystems form. Microbial communities play important roles in these aquatic ecosystems and can be identified using environmental DNA (eDNA) acquisition and amplification techniques. The Spring Mountain range west of Las Vegas, Nevada is a sky island with over 300 naturally flowing cold water springs. eDNA was acquired at three sampling sites at two springs from different elevations on the eastern side of the range–Harris Spring (HS) located at 1,779 m elevation and Deer Creek Spring (DCS) located at 2,778 m elevation–to compare microbial community composition, abundance, alpha diversity, and median evenness. Aquatic chemistry parameters such as water temperature, pH, conductivity, total dissolved solids (TDS), sodicity, alkalinity, and dissolved oxygen were measured to quantify environmental conditions within the springs. Results show that DCS exhibited greater microbial abundance, alpha diversity, and median evenness when compared to HS. This differs from elevational diversity gradient expectations which predicts lower abundance, alpha diversity, and median evenness due to harsher environmental conditions and scarcer resources. Aquatic chemistry measurements indicated no significant difference in pH and dissolved oxygen between the two springs, and differences in water temperature and conductivity do not provide a substantive explanation for microbial community patterns observed. A discussion of other environmental variables that were observed to qualitatively differ between the study sites, such as exposure to solar radiation, stream reemergence, and anthropogenic disturbance, are offered as potential explanatory variables that warrant further investigation. Further research is needed to better quantify these relationships, which would also assist in the environmental conservation efforts of these fragile spring ecosystems.

**Data availability statement:** All relevant data are within the paper. DNA sequencing results (FASTQ files) can be found on the figshare repository with the following DOI: https://doi.org/10.6084/m9.figshare.30715514. This DOI is live and all 13.7 GB of data are available at this repository.

**Funding:** Funding was made possible by a grant from the National Institute of General Medical Sciences (GM103440) from the National Institutes of Health through the Nevada Idea Network of Biomedical Research Excellence (INBRE) grants for both faculty and undergraduate students, Regional Alliance of INBRE Networks (RAIN) Technology Access Grants, and Nevada INBRE Scientific Core Services Awards.

**Competing interests:** NO authors have competing interests.

## Introduction

Springs are ecotones: where groundwater intersects at or nearly with topographic elevation and creates localized ecosystems based on surface water availability [1,2]. [1] identified twelve types of springs, termed "spheres of discharge," based on surface expression of water flow (source emergence), landscape morphology, and presence of plant and animal species. Springs create microhabitats that can support diverse and unique organisms not found elsewhere in the region [2]. Cold water springs are those which have not been geothermally-heated by Earth's internal thermal processes (e.g., proximity to magmatic heat source) and with source water that has remained at naturally-cool surficial temperatures, typically equal to the mean air temperature of a region for the entirety of its groundwater residence. Cold springs typically exhibit marginal fluctuations in physical and chemical properties [3].

The Spring Mountains are located west of Las Vegas in Clark County, Nevada. This mountain range includes the 3,633-m Charleston Peak (also known as Mount Charleston) among many other peaks with an elevation greater than 3,048 m above sea level (e.g., Fletcher Peak, Griffith Peak, McFarland Peak, and Mummy Mountain). Charleston Peak is the nineteenth most prominent summit in the United States with a prominence of 2,517 m. Surrounded on all sides by the Mojave Desert and with such great elevations, prominence, and an isolation of 220 km to the closest mountains reaching similar heights, the Spring Mountains are considered a sky island, as described by [4,5]. In addition to spatial isolation, the Spring Mountains have experienced temporal and climatic isolation: in the early Holocene, the mountains were surrounded by valley-wide wetlands and springs and since about 6,000 years ago have been surrounded by arid environments. This temporal isolation for thousands of years has led to the evolution of endemic species, including small mammals, mollusks, insects, and plants [4–7].

The Spring Mountains were named for the over 300 springs that emerge and flow at all elevations of the mountain range and provide resources for approximately 80% of terrestrial animal species [4,8]. Spring Mountains springs tend to exhibit peak discharge in late spring and decline to base flow by fall or winter months, reflecting a pattern of snowmelt recharge feeding the springs [9,10]. Groundwater residence time and flow path from snowmelt to spring source location is not well-understood due to complicated bedrock geology of highly fractured yet consolidated carbonate rocks exhibiting low primary porosity but high secondary porosity [9,11,12], so understanding groundwater flow path requires detailed subsurface mapping of fractures and faults. Additionally, many springs, particularly lower-elevation springs, have been altered by human activity [8] like tourism and recreation, small community development, historic mining, and the introduction of mammals (e.g., horses and burros) that trample and overgraze the springs [4]. Typically, lower-elevation springs are more accessible (i.e., closer to roads, shorter hiking trails, on flatter terrain, etc.) and show evidence of being more altered than higher-elevation springs. [8] found "the response of many aquatic taxa in the Spring Mountains to disturbances is similar regardless of the source…" and showed species richness to be greater at mildly disturbed springs than highly disturbed springs.

Springs are considered one of the most threatened ecosystem types [1,2]. This is especially the case in the southwestern United States, where some regions have experienced drought conditions since the year 2000 [13,14]. Due to impending threats of disturbance and drought, many researchers are applying eDNA technologies to study these springs before they are lost, e.g., [5,8,15–20].

The purpose of this study is to contribute to this knowledge base through a microbial survey of two different springs (Harris Spring and Deer Creek Spring) located within a sky island ecosystem surrounded by the Mojave Desert in southern Nevada. The goals of this study are to (1) inventory microbial communities, (2) compare microbial communities between the springs, and (3) discuss possible causes of observed differences between microbial community composition due to external environmental parameters (e.g., water chemistry, solar radiation, climate, disturbance levels, etc.). Currently, no cold-water springs on the eastern slopes of the Spring Mountains in southern Nevada have been studied in such a manner, and these springs act as an alpine groundwater source that eventually flows into the Las Vegas Valley watershed [9,11].

## Materials and methodology

### Study sites

Two springs on the eastern side of the Spring Mountains, included in the Las Vegas Valley Watershed [9,21], were selected to study microbial community composition: the lower-elevation Harris Spring (HS) and the higher-elevation Deer Creek Spring (DCS)--see S1 Table and project workflow in Fig 1. HS is located along Harris Spring Road on the south side of Kyle Canyon Road at 1,179 m above sea level. HS emerges in a small valley with 4–6-m walls adjacent to a shared community space with approximately ten buildings. It flows as a rheocrene spring (i.e., water emerges from the ground and flows in a channel) over relatively flat and wide terrain for 1.6–3.2 km depending on the season. This spring experiences frequent disturbances, both from large ungulates grazing and trampling the spring and by humans recreating around and within the spring (i.e., driving vehicles through the stream channel and dumping trash near the spring source). DCS emerges from a steep slope in the forest and flows as a narrow rheocrene spring about 1.6 km while descending over 305 m of elevation. In contrast to HS, DCS only experiences a high level of disturbance near its base–which intersects the Deer Creek Picnic Area and the main road connecting Kyle Canyon and Lee Canyon–but experiences much less disturbance near its source and upstream reaches. Historically, both springs experienced some disturbance through small-scale infrastructure development and installation of near-surface metal pipes and cisterns for water diversion and collection.

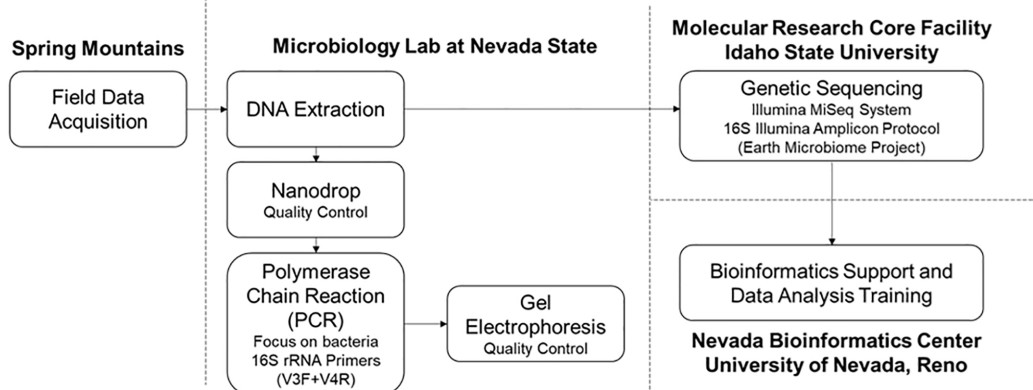

**Fig 1. Project workflow.**

Three sites were sampled at Harris Spring (HS) and Deer Creek Spring (DCS): (1) Upper (-U) for the site at or most proximal to the spring source, (2) Middle (-M) for the next site downstream from the source, and (3) Lower (-L) for the site farthest downstream from the source. Site names and abbreviations, locations, and brief descriptions are shown in Table 1. The three HS sites were located within the high anthropogenic disturbance areas, while the three DCS sites were located well upstream from the high anthropogenic disturbance areas. All sites at both springs were sampled on three different dates (S1 and S2 Tables) throughout June and July 2022.

## eDNA acquisition

DNA released into the environment by organisms is collectively known as environmental DNA (eDNA). Protocols to collect, extract, amplify, and sequence eDNA have been developed based on various target organisms (Fig 1). Advances in high-throughput technologies have allowed for community-level analyses (e.g., hundreds of thousands to millions of eDNA fragments if present per sample) of target organisms [22–24].

eDNA samples were collected on the dates described in S1 Table and at spring locations described in Table 1. Sample collectors wore nitrile gloves disinfected with a 70% ethanol solution. Five liters of spring water was passed through a Millipore® Sterivex™ 22-μm luer-lock pressure filter using a sterilized 60 mL luer-lock syringe and was returned to the spring (similar methodology as previous studies, i.e., [3,25,26]). No spring water was collected or removed from the sampling sites. Permits were not required as both study sites were located on public lands, and neither water samples nor protected species were removed from the sampling locations. The filters were dried (excess water removed to avoid cracks formation in the glass during freezing), capped, labeled, stored in a sterilized plastic bag, and placed in a small cooler on dry ice. A control sample was also collected once per field day, typically at the upper location, which involved passing air through a filter to account for any microbial contamination from the sample collector. All filters were transported on dry ice and placed in a −20°C freezer for preservation (i.e., [27]) for less than one week prior to further laboratory procedures (as described below).

**Table 1. Upper, middle, and lower site information at Harris Spring and Deer Creek Spring.**

**Harris Spring**

|  | Upper (HS-U) | Middle (HS-M) | Lower (HS-L) |
|---|---|---|---|
| Latitude | 36.23994 | 36.24054 | 36.24109 |
| Longitude | −115.54392 | −115.54354 | −115.54292 |
| Elevation | 1,783 m | 1,780 m | 1,774 m |
| Description | Rheocrene stream (about 3 m downstream from source) is contained in a valley with steep sediment walls up to 3–4.5 m high; consistent water flow rates; frequent disturbance; ranges in depth 10–20 cm and width 20–51 cm. | Rheocrene stream with ponding; highly dynamic water flow rates (less than upper and lower); frequent disturbance; up to 30–46 cm in depth and 1.8–2.4 m in width. | Rheocrene stream; highly dynamic water flow rates; frequent disturbance; ranges in depth 2.5–10 cm and width 30–60 cm. |

**Deer Creek Spring**

|  | Upper (DCS-U) | Middle (DCS-M) | Lower (DCS-L) |
|---|---|---|---|
| Latitude | 36.30677 | 36.30683 | 36.30679 |
| Longitude | −115.63745 | −115.63624 | −115.63352 |
| Elevation | 2,835 m | 2,793 m | 2,732 m |
| Description | Rheocrene stream and hillslope spring emergence; consistent water flow rates; low disturbance; ranges in depth 5–10 cm and width 15–30 cm. | Rheocrene stream; consistent water flow rates; infrequent disturbances; ranges in depth 10–20 cm and width 20–46 cm. | Rheocrene stream; consistent water flow rates; infrequent disturbance; ranges in depth 10–30 cm and width 30–61 cm. |

## Aquatic chemistry

Environmental conditions were monitored simultaneously and downstream adjacent to eDNA acquisition locations by a field crew supervised by the authors. Oakton Instruments PCTSTestr5 (WD-35634–60) handheld water meters were used to directly measure water temperature, pH, and electrical conductivity after they were properly calibrated using three pH standards (4.01, 7.00, and 10.01) and one conductivity standard (1,413 µS/cm). The PCTSTestr5 water meters were used to also provide estimates for total dissolved solid (TDS) and sodicity (S) concentrations:

$$TDS = 0.7\sigma \text{ and } S = 0.5\sigma$$

where σ is electrical conductivity (µS/cm). Values for TDS and S are indirect calculations based entirely on electrical conductivity. Additionally, S is referred to as sodicity here–even though the product is advertised to "measure salinity"--communication with the manufacturer revealed that the indirect calculations for S were based on assumptions of only sodium chloride (NaCl) concentrations (sodicity) and not all dissolved salts (salinity). Two other variables were measured using visual color-based test kits: alkalinity using the Fisher Scientific Total Alkalinity Test Kit and dissolved oxygen concentration using the CHEMetrics Dissolved Oxygen Test Kit.

Water chemistry parameters (temperature, pH, conductivity, TDS, S, alkalinity, and dissolved oxygen) were compared between HS and DCS using mixed-effects models. For each parameter, Spring was treated as a fixed effect, while Site nested within Spring was included as a random effect to account for repeated measures at different locations within each spring. Models were fitted using the lme4/lmerTest package in R (version 4.5.2; [28]), and statistical significance of the Spring effect was assessed using likelihood ratio tests comparing the full model with a null model containing only the random effects. Missing values (N/A) for alkalinity and dissolved oxygen were retained in the analysis. Parameters with insufficient observations (<6 non-missing values) were excluded from testing. Significance was considered at $\alpha = 0.05$.

## Laboratory procedures

Laboratory protocols followed the Earth Microbiome Project (EMP) 16S Illumina Amplicon Protocol which amplifies prokaryotic (bacteria and archaea) DNA using paired-end 16S community sequencing on the Illumina platform with "[p]rimers 515F-806R [to] target the V4 region of the 16S SSU rRNA [29]. This EMP protocol is considered a standard approach for working with eDNA and is supported by several studies [30–36]. Laboratory procedures specific to this project are provided below.

The extraction of eDNA was performed in a research laboratory at Nevada State University using the Qiagen DNeasy PowerWater Sterivex Kit according to the kit protocol. Extracted DNA was packaged and shipped on dry ice for sequencing at The Molecular Research Core Facility at Idaho State University using the 16S Illumina Amplicon Protocol described in the EMP 16S Illumina Amplicon Protocol. Single libraries were generated for each sample targeting the 16S rRNA gene. Amplicons were created following the [29] utilizing the V4 prokaryote primers. Library integrity was assessed using Agilent Fragment Analyzer and qPCR for quality control. The library pool was sequenced on an Illumina v2 500-cycle sequencing kit and produced FASTQ data files (see S1 File). The Nevada Bioinformatics Center at University of Nevada, Reno performed statistical analyses and bioinformatics on the FASTQ files to identify patterns and trends in overall microbial community composition and diversity.

## Results

Aquatic chemistry measurements were acquired at HS and DCS on three different dates approximately two weeks apart (S1 Table) during summer 2022. Both springs exhibited alkaline water (pH > 7), with DCS reporting slightly higher but not statistically different pH level compared to HS (Table 2 and S2 Table). The water temperature was markedly higher at HS, and conductivity, TDS, and salinity were 2–3 times greater at HS compared to DCS. HS also exhibited greater alkalinity (ppm $CaCO_3$) and slightly lower dissolved but not significantly different oxygen levels (Table 2 and S2 Table).

**Table 2. Comparison of water chemistry parameters between HS and DCS.** Means±SD were calculated across all sites and sampling dates within each spring. F-values (numerator df=1, denominator df≈16) and p-values were obtained from mixed-effects ANOVA models with Spring as a fixed effect and Site nested within Spring as a random effect. Significant differences (p<0.05) are shown in bold.

| Parameter | Harris Spring (Mean±SD) | Deer Creek Spring (Mean±SD) | F(1,16) | p-value |
|---|---|---|---|---|
| Water Temperature (°C) | 16.0±2.2 | 8.9±3.3 | 5.40 | **0.020** |
| pH | 7.95±0.31 | 8.19±0.21 | 2.02 | 0.155 |
| Conductivity (µS/cm) | 697±37 | 288±57 | 11.30 | **0.001** |
| Total Dissolved Solids (ppm) | 492±25 | 188±40 | 11.26 | **0.001** |
| Salinity (ppt NaCl) | 0.35±0.02 | 0.14±0.04 | 11.24 | **0.001** |
| Alkalinity (ppm CaCO3) | 297±66 | 117±54 | 4.87 | **0.027** |
| Dissolved Oxygen (ppm) | 7.0±3.5 | 7.3±8.0 | 1.94 | 0.163 |

Rarefaction curves of the microbial communities for all samples acquired are shown in Fig 2. The number of species per sample ranges from ~1,250 to almost 10,000. All rarefaction curves reached an asymptote, an indication of ample sample depth. In general, more species were identified at DCS than at HS, DCS-L had the most species, and HS-M and HS-U had the fewest species.

Fig 3 displays box and whisker plots showing the Inverse Simpson Index (1/D) for samples acquired at DCS (left) and HS (right) at each location (lower, middle, and upper). 1/D is directly proportional to evenness, which is a metric of how species are distributed within a community. Thus, a low 1/D indicates a single dominant species while a high 1/D implies an evenly distributed community with like-number of prevalent species. Theoretically, the lower bounds of 1/D=1 and the upper bounds is S (number of species), where 1/D=1 means all individuals are of one species and 1/D=S means there is one individual present from every species.

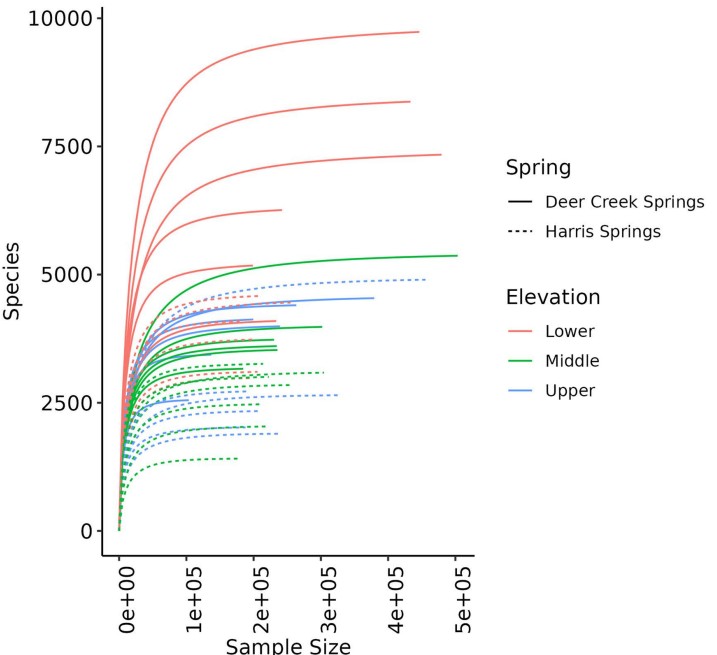

**Fig 2. Rarefaction curves for eDNA acquired at all three locations (lower, middle, and upper) at HS and DCS.** Does not include controls (filters pumped with air).

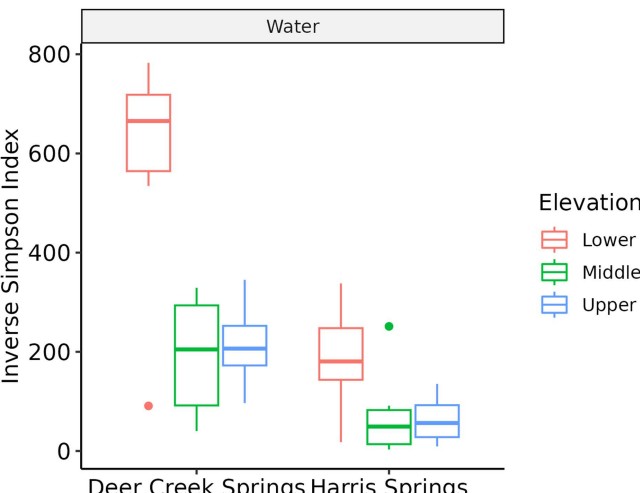

**Fig 3. Box and whisker plots showing inverse Simpson index for all locations at HS and DCS.**

Spring locations can be categorized into three evenness groups: DCS-L (1/D~700); DCS-M, DCS-U, and HS-L (1/D~200); HS-M and HS-U (1/D~50). Overall, DCS exhibited greater evenness than HS.

A non-metric dimensional scaling (NMDS) plot collapses multivariate data into two dimensions. The Bray-Curtis method was used in the creation of these NMDS plots (Figs 4 and 5) as it accounts for rank orders of species presence and abundance, among other variables. In short, the NMDS procedure is iterative and generally follows these steps: (1) define starting positions of communities in the original multidimensional space, (2) configure all samples into a collapsed n-dimension (typically two) based on specified starting conditions, (3) regress starting positions of all samples against measured positions, (4) calculate stress, which is the disagreement level between regressed and observed positions, and,

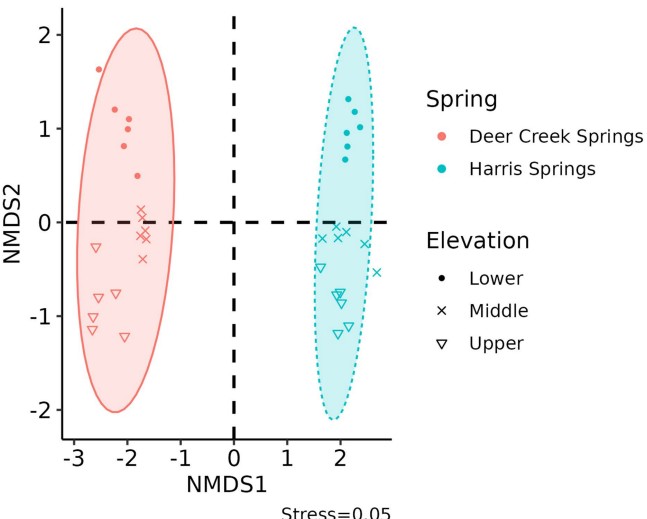

**Fig 4. Non-metric multidimensional scaling plot with 95% confidence ellipses separated by spring (DCS as red; HS as blue) and location (lower with a circle, middle with an x, and upper with a triangle).**

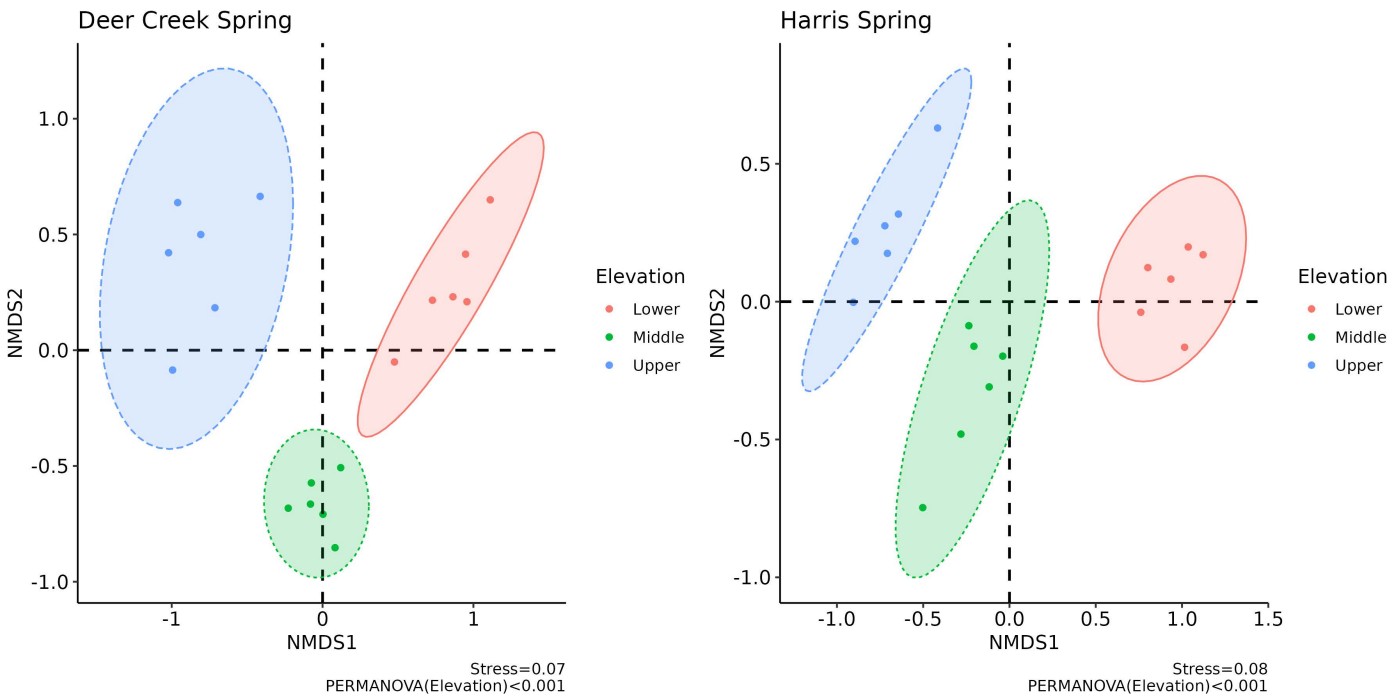

**Fig 5. Non-metric multidimensional scaling plots with 95% confidence ellipses separated by location (lower, middle, and upper) at HS and DCS.**

if stress is high, reposition towards a condition with lower stress and reiterate. A low stress value indicates agreement between NMDS positions (locations of points on the plot) and observations measured in the field. Figs 4 and 5 show final stress values of 0.05 and 0.07–0.08, respectively.

Microbial community differences were observed between HS and DCS and within each spring, as shown in Fig 4. The NMDS1 dimension highlights complete dissimilarity in microbial community compositions between DC and HS due to the lack of 95% confidence ellipses overlap across this axis. The NMDS2 dimension shows a distinction in microbial community compositions within each spring through the separation of location (i.e., elevation) point clusters across this axis. Microbial community compositions differed more between springs (NMDS1) than within springs (NMDS2).

A NMDS plot was created for each spring (Fig 5) and shows microbial community composition comparisons by location (i.e., elevation). Both springs exhibited distinct dissimilarities across their respective locations in the NMDS1 dimension. However, both springs also exhibited similarities across their respective locations in the NMDS2 dimension. These results illustrate the complexity and changes to microbial community composition along the rheocrene spring profiles.

Fig 6 shows absolute abundances of ten major phylum groupings and "others" at each spring and location. The x-axis of this plot shows sample names categorized by location, and the y-axis shows absolute abundances (count) by spring. Overall, the most abundant phylum present in both springs is proteobacteria. Cyanobacteria is the second most abundant at HS. Bacteroidota is the second most abundant at DCS.

## Discussion

Microbial community composition differs at two geographical scales. The larger-scale geographic separation between HS and DCS, 23 km distance by roads and trails climbing 1,000 m in elevation, yields greater differences in community composition than the smaller-scale geographic separation within each spring. This finding is supported by many previous

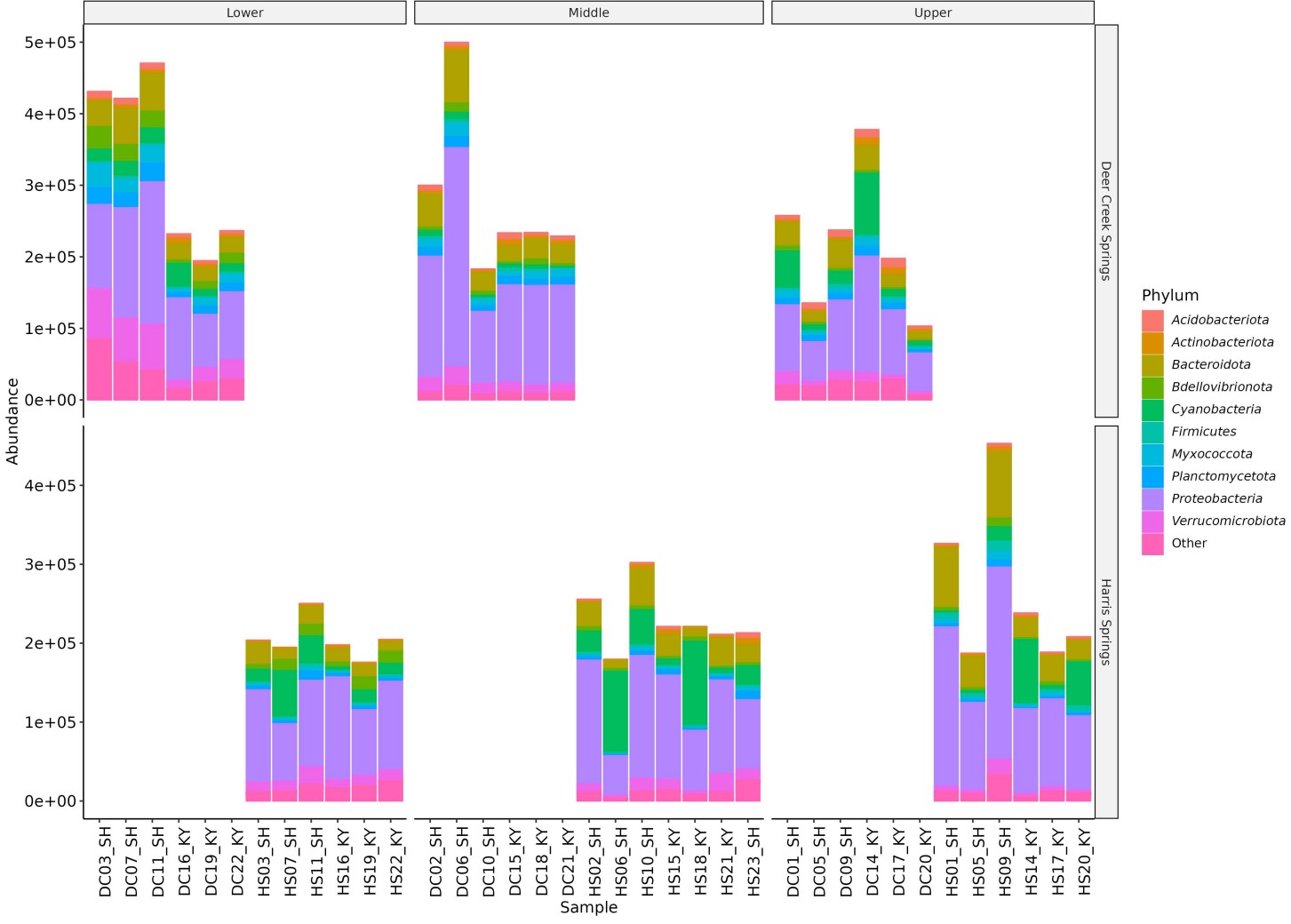

**Fig 6.  Absolute abundance of bacteria and archaea by phylum found at each location at each spring (HS refers to Harris Spring and DC refers to Deer Creek in the sample names).**

studies, which have shown environmental factors, like aquatic chemistry, play a role in microbial community structure and diversity in springs (i.e., [3]). Results from this study show a significant difference in water temperature, conductivity, TDS, sodicity, and alkalinity between HS and DCS (Table 2). This, among other variables discussed below, may play an important role in the distinct separation of 3–4 units on the NMDS1 axis (Fig 4) exemplifying just how different microbial communities are across the larger-scale geographic separation within the same sky island ecosystem.

Species abundance and median evenness within the microbial community at DCS was greater than that measured at HS (Fig 6). This was surprising because the expectation was that these metrics would adhere to the elevational diversity gradient, which states that species count decreases with elevation due to harsher environmental factors and resource limitations [37]. The microbial community at DCS also exhibited a greater median evenness (1/D~200–700) indicating many more species were present at roughly equivalent abundance compared to HS (1/D~50–200) suggesting the presence of fewer dominant species (Fig 3). The most abundant phyla at DCS were proteobacteria, bacteroidota, and cyanobacteria, while the most abundant phyla at HS were proteobacteria, cyanobacteria, and bacteroidota (Fig 6).

## Elevational diversity gradient

The findings that diversity was greater in the higher elevation site (DCS) ran counter to the well-described elevational diversity gradient. Some data and qualitative observations provide potential explanations as to why DCS has greater species abundance and median evenness compared to HS and may warrant further investigation. An exploratory list of variables and brief explanations as to how these variables may impact microbial community composition are provided below.

**Aquatic chemistry.** Differences in quantitative measurements of aquatic chemistry between DCS and HS for variables like water temperature, conductivity, TDS, sodicity, and alkalinity offer a potential explanation for differences in the microbial community. Observed differences in aquatic chemistry parameters in this study match general expectations in alpine environments: water temperature decreases at higher elevations and conductivity/TDS/sodicity tends to decrease as elevation gradient increases (due to shorter contact times between water and rocks/soils to allow for dissolution of minerals). Also, the lack of difference in dissolved oxygen and pH between the sites implies that these variables were not the driving factors for the differences in the microbial community. However, there is no indication in previous studies that any of the measured changes in aquatic chemistry values would alone account for the microbial community differences between DCS and HS.

**Solar radiation.** Exposure to solar radiation can directly impact microbial abundance, microbial evenness, and community composition. Radiation damages DNA and microbes with radiation-resistant traits may survive prolonged exposure compared to microbes lacking those traits [38–40]. DCS is mostly shaded by the mixed forest of Douglas fir, spruce, and aspen. It is also located on an east-facing slope and does not get direct sunlight in the late afternoon and evenings. HS is entirely exposed to direct sunlight for most of the day, with little shade in the morning and evening from distant mountain peaks. This difference in solar radiation exposure could be a plausible reason why DCS has greater abundance, alpha diversity, and species evenness compared to HS.

**Anthropogenic disturbances.** Humans have historically disturbed both springs. A popular picnic area exists where DCS flows in a culvert beneath highway 158 which connects Kyle Canyon and Lee Canyon. To reach DCS-L from the picnic area one must hike ~3 km and climb over 300 m, including over 1 km of off-trail hiking. Only the occasional hikers were seen during field investigations at DCS, although historical evidence of human presence in the area was noticeable (i.e., installation of metal pipes over 50 years ago to divert water to homes down the slope). Humans heavily impacted HS due to accessibility to the site: anyone can drive the ~5 km dirt road and park their car *in* the spring. There is a housing complex adjacent to HS-U, and human presence was repeatedly noted during field investigations. There were also off-road vehicle tracks in the rheocrene stream channel at HS-L. Previous studies have noted the impacts of anthropogenic disturbances on soil and water microbial communities [41–43] and indicate human impacts may outweigh natural disturbances [44].

**Natural stream reemergence.** Abundance and persistence of eDNA varies significantly by medium, typically highest in sediment and soil, lower in water, and lowest in air (i.e., [45,46]). Both DCS and HS are rheocrene springs and flow on channelized stream beds. Water flowing in DCS submerged and reappeared multiple times between DCS-M and DCS-L. Repeated submergence-reemergence may have contributed to greater abundance and alpha diversity through the accumulation of eDNA from soil and sediment when compared to HS, which did not display such natural stream reemergence (i.e., HS water emerged from its groundwater source upstream of HS-U and remained as surface water through HS-M and HS-L).

The Spring Mountains are a natural sky island ecosystem surrounded by the arid Mojave Desert. These springs are the primary water resource for many animals living in and around the riparian environments they create. Utilizing the elevational diversity gradient, a comparison of the differences between DCS and HS microbial community composition, abundance, alpha diversity, and median evenness could yield two assumed outcomes: (1) DCS microbial community variables match expected elevational diversity gradient values and HS microbial community variables are abnormally low, or (2) HS microbial community variables match expected elevational diversity gradient values and DCS microbial community

variables are abnormally high. Based on the quantitative measurements of aquatic chemistry and qualitative observations of solar radiation, anthropogenic disturbance levels, and natural stream reemergence, we believe the more likely explanation is that the community patterns described in the study are mainly due to exposure to solar radiation and relatively high human disturbance levels at HS. Further research is needed to quantify these environmental and anthropogenic variables to better understand the degree of impact they may have on microbial community composition. Results from further studies may also aid regional governmental agencies and nonprofits that collaborate to "survey, rehabilitate, and steward spring systems" [47] in the southwestern United States.

## Conclusion

Aquatic chemistry and microbial communities were studied at two naturally flowing cold-water springs located at significantly different elevations in the Spring Mountains west of Las Vegas, Nevada. Deer Creek Spring (the higher elevation spring) was colder, more alkaline, had lower conductivity and TDS/salinity concentrations, exhibited a greater absolute abundance of microbial species, and a higher evenness within the microbial community compared with Harris Spring (the lower elevation spring). Potential explanations for these differences include elevation, topography (which impacts climate and precipitation), and ecosystem, but anthropogenic disturbance and direct solar radiation may be important factors as to why Deer Creek Spring unexpectedly has a more robust and even microbial community than Harris Spring and warrants additional investigation.

This paper provides a preliminary survey of the microbial community of cold-water springs on an understudied sky island and the unique Spring Mountains ecosystem. Abundant additional research opportunities are available at Harris Spring, Deer Creek Spring, and other cold-water springs of the Spring Mountains in the areas of microbial biology (e.g., detailed bioinformatic analysis on microbial communities, discovery of novel endemic bacteria that can be used in future antibiotics, etc.), hydrology (e.g., flow path of the springs in three dimensions), geochemistry (e.g., how soil and bedrock impact aquatic chemistry parameters), and climate change (e.g., how these delicate spring ecosystems change over time). Additionally, there are ~300 more springs that can be studied, each with their own unique challenges, including anthropogenic and large mammal disturbances, presence of endemic species, variable water levels throughout seasons due to climate change, etc. The Spring Mountains provide a unique outdoor laboratory that is easily accessible near a large urban area that has been relatively understudied. The study also highlights the need for conservation and management efforts to protect the unique freshwater springs of these sky island habitats, especially if anthropogenic disturbance of the springs is significantly impacting their microbial communities.

## Supporting information

**S1 Table. Characteristics of Harris Spring and Deer Creek Spring.**
(DOCX)

**S2 Table. Water chemistry parameters by measurement date at Harris Spring and Deer Creek Spring.**
(DOCX)

**S1 File. FASTQ files for Spring Mountains, Nevada.**
(DOCX)

## Acknowledgments

The authors would like to acknowledge the agencies, labs, and individuals who assisted on this project, including the National Institutes of Health, the Idaho State University Molecular Research Core Facility, and the Nevada Bioinformatics Center. The authors would like to thank the Institutes of Health through the Nevada Idea Network of Biomedical Research Excellence (INBRE) summer field students who helped pump liters of water through Sterivex filters.

## Author contributions

**Conceptualization:** Shawna Hunnicutt, El Hachemi Bouali.

**Data curation:** Shawna Hunnicutt, El Hachemi Bouali.

**Formal analysis:** El Hachemi Bouali, Bryan Sigel.

**Funding acquisition:** El Hachemi Bouali, Bryan Sigel.

**Investigation:** Shawna Hunnicutt, El Hachemi Bouali, Bryan Sigel.

**Methodology:** Shawna Hunnicutt.

**Project administration:** El Hachemi Bouali, Bryan Sigel.

**Resources:** Bryan Sigel.

**Supervision:** El Hachemi Bouali.

**Validation:** Bryan Sigel.

**Visualization:** Bryan Sigel.

**Writing – original draft:** El Hachemi Bouali.

**Writing – review & editing:** Bryan Sigel.

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
