## [Decision Letter · Decision Letter 0]

2 Oct 2025

Dear Dr. Bouali,

Thank you for submitting your manuscript to PLOS ONE. After careful consideration, we feel that it has merit but does not fully meet PLOS ONE’s publication criteria as it currently stands. Therefore, we invite you to submit a revised version of the manuscript that addresses the points raised during the review process.

We look forward to receiving your revised manuscript.

Kind regards,

Renjith VishnuRadhan, PhD

Academic Editor

PLOS ONE

Journal Requirements:

“Funding was made possible by a grant from the National Institute of General Medical Sciences (GM103440) from the National Institutes of Health through the Nevada Idea Network of Biomedical Research Excellence (INBRE) grants for both faculty and undergraduate students, Regional Alliance of INBRE Networks (RAIN) Technology Access Grants, and Nevada INBRE Scientific Core Services Awards.”

“The authors would like to acknowledge the agencies, labs, and individuals who assisted on this project. Funding was made possible by a grant from the National Institute of General Medical Sciences (GM103440) from the National Institutes of Health through the Nevada Idea Network of Biomedical Research Excellence (INBRE) grants for both faculty and undergraduate students, Regional Alliance of INBRE Networks (RAIN) Technology Access Grants, and Nevada INBRE Scientific Core Services Awards. Genomic sequencing, metabarcoding, and library  creation were completed by Idaho State University Molecular Research Core Facility, RRID:SCR_012598 (https://www.isu.edu/mrcf/). Statistical analyses and bioinformatics were completed by the Nevada Bioinformatics Center (https://www.unr.edu/bioinformatics). Lastly, the authors would like to thank the INBRE summer field students who helped pump liters of water through Sterivex filters.”

“Funding was made possible by a grant from the National Institute of General Medical Sciences (GM103440) from the National Institutes of Health through the Nevada Idea Network of Biomedical Research Excellence (INBRE) grants for both faculty and undergraduate students, Regional Alliance of INBRE Networks (RAIN) Technology Access Grants, and Nevada INBRE Scientific Core Services Awards.”

5. We note that you have indicated that there are restrictions to data sharing for this study. PLOS only allows data to be available upon request if there are legal or ethical restrictions on sharing data publicly. For more information on unacceptable data access restrictions, please see http://journals.plos.org/plosone/s/data-availability#loc-unacceptable-data-access-restrictions.

6. We note that Figure 2b includes an image of a participant.

Please respond by return e-mail with an amended manuscript. We can upload this to your submission on your behalf.

If you are unable to obtain consent from the subject of the photograph, please either instruct us to remove the figure or supply a replacement figure by return e-mail for which you hold the relevant copyright permissions and subject consents. In some cases, you may need to specify in the text that the image used in the figure is not the original image used in the study, but a similar image used for illustrative purposes only. We can make any changes on your behalf.

Reviewers' comments:

Reviewer's Responses to Questions

**Comments to the Author**

1. Is the manuscript technically sound, and do the data support the conclusions?

Reviewer #1: No

Reviewer #2: Partly

Reviewer #3: Yes

Reviewer #4: Yes

2. Has the statistical analysis been performed appropriately and rigorously?

Reviewer #1: No

Reviewer #2: No

Reviewer #3: Yes

Reviewer #4: N/A

3. Have the authors made all data underlying the findings in their manuscript fully available?

Reviewer #1: No

Reviewer #2: Yes

Reviewer #3: Yes

Reviewer #4: Yes

4. Is the manuscript presented in an intelligible fashion and written in standard English?

Reviewer #1: No

Reviewer #2: Yes

Reviewer #3: Yes

Reviewer #4: Yes

Reviewer #1: PONE-D-25-05980

A comparison of the aquatic chemistry and microbial community of two cold-water springs in the Spring Mountains of southern Nevada

This manuscript is not ready for publication in a scientific journal. The report appears as an undergraduate project and reads that way. Although the topic of bacteria in oligotrophic waters can be of importance and interest, this project appeared to be the training of students in both engaging students in field work as well as training them on bioinformatic and eDNA, and less about sound hypothesis development and robust science to answer the questions.

Data are not yet available (see front matter).

Please see some constructive comments below:

Title: Use of spring twice in the title might be reconsidered.

Abs:

1. Sky island is likely not a geographic term.

2. Mention how many sites per spring were sampled.

3. What is your hypothesis?

4. What is the scientific importance/justification for the study?

5. Write in past tense (throughout).

6. The last sentence is an assumption.

Intro:

7. Reads like a book report. (Authors cite page numbers of others work). What is the relevance of this material to your study?

8. What is your hypothesis and the justification for this study?

9. Line 53-59. Define metrics, water chemistry, and microbiome.(Ln. 149 – be specific)

10. Be more thorough about description of eDNA sources (Line 60).

11. Ln. 64. How can eDNA be in samples?

12. Suggest making a table for background on eDNA applications with citing specific articles relevant only to your study. Lns 66-69 are not comprehensive.

13. Ln. 74. Trending means what?

14. Ln 83. Sky island?

15. Ln. 91. That 90% figure cannot be correct.

16. Ln. 101. “Wild” is inappropriate.

17. Ln. 110. Only now is community composition discussed and although it is a focus for the survey, it seems like an afterthought in the way it is being present. Ln.

18. Rheocrene and similar terms need definition and citation.

19. There is little rationale for site selection along the Springs.

20. Tables. Should be in supplement. Why data in the Intro?

21. Ln. 152. Microorganisms can in fact travel by air – provide context and citations.

Methods:

22. Throughout there is not enough information on instruments for manufacturer, sources, and details.

23. Many details can be included in supplement.

24. Ln. 188. Water samples from which eDNA was derived were collected. Correct the wording, please.

25. Ln. 192. That high number (83 x) of filtering with a 60 mL syringe with a filter is not realistic or practical. The number of times is not a competition (see Ln. 193). Errors and contamination could be introduced.

26. Ln. 201. Cite the use of the -20C rather than -80C.

27. Ln. 215. Who did the work? Specify.

Results:

28. Results. Present them, not generalities.

29. Show the data in interesting way, then use the discussion to interpret the data.

30. Ln. 244. Opportunistic measures should have been described in the methods, with that rationale.

Discussion:

31. Discuss your own data in relations to others. For example, Ln. 424. Did you measure UV or any other metric such as landcover that is a less subjective than an observation of tree species nearby.

Figures 1 and 2 are not very helpful.

Figure legends not near figures is cumbersome. Axes labels need explanation.

Reviewer #2: The statement “Water temperature was higher at Harris Spring than Deer Creek Spring” in line 229 is indefinite as the actual temperature figures were not mentioned whereas succinct description is the results is main goal of this section.

Results have been presented in bullet points like power point presentation.

ANOVA should be conducted to statistically compare the physicochemical parameters across the stations before a comparison can be made with declaration of the probability level. Authors compared activities around the catchment area and this is not the right approach.

Reviewer #3: Manuscript ID: PONE-D-25-05980

Title: A comparison of the aquatic chemistry and microbial community of two cold-water springs in the Spring Mountains of southern Nevada

Overall Evaluation

The manuscript is well-written, methodologically sound, and provides valuable insights into the chemistry and microbial community composition of cold-water springs in the Spring Mountains. The integration of environmental DNA (eDNA) with aquatic chemistry strengthens the work and fills a regional knowledge gap. The findings on elevation, disturbance, and microbial diversity are novel and important for conservation and water management in arid landscapes.

Points for Improvement

The introduction can be strengthened by linking spring ecosystem studies to broader hydrologic and climatic processes. For instance, hydropower and reservoir impacts on hydrologic cycles (Li et al., 2025; Huang et al., 2025) and global isotope datasets (Li et al., 2025) provide useful comparisons.

The discussion mentions drought but could be deepened by linking with global precipitation extremes and warming–hydrology interactions (Zhang & Wu, 2025; Zhu et al., 2024).

The geomorphology section could draw from case studies in other volcanic and fractured terrains (D Silva et al., 2024; Abegeja & Nedaw, 2024). This would highlight how fractured geology and landscape evolution influence spring chemistry.

The role of anthropogenic disturbance is well described. Linking this with applied remediation or bio-inspired solutions (Ma et al., 2025) could increase applied relevance.

Minor Comments

• Figures are clear, but NMDS plots could include 95% confidence ellipses.

• Methods should clarify whether “disturbance level” was quantified or based on qualitative observations.

• Conclusion should explicitly highlight conservation and management implications for cold-water springs in arid mountain ecosystems.

Recommendation

Minor Revision – The study is publishable after addressing the contextual gaps in the introduction/discussion and incorporating broader hydrological and climatic perspectives.

Suggested References to Add

• Li, R., Zhu, G., Lu, S., Meng, G., Chen, L., Wang, Y.,... Wang, Q. (2025). Effects of cascade hydropower stations on hydrologic cycle in Xiying river basin, a runoff in Qilian mountain. Journal of Hydrology, 646, 132342. https://doi.org/10.1016/j.jhydrol.2024.132342

• Huang, E., Zhu, G., Meng, G., Wang, Y., Chen, L., Miao, Y.,... Li, W. (2025). Historical dataset of reservoir construction in arid regions. Scientific Data, 12(1), 1428. https://doi.org/10.1038/s41597-025-05712-3

• Yi, Z., Qiu, C., Wang, D., Cai, Z., Yu, J.,... Shi, J. (2024). Submesoscale kinetic energy induced by vertical buoyancy fluxes during the tropical cyclone Haitang. JGR: Oceans, 129(7), e2023JC020494. https://doi.org/10.1029/2023JC020494

• Li, R., Zhu, G., Chen, L., Qi, X., Lu, S., Meng, G.,... Gun, Y. (2025). Global stable isotope dataset for surface water. Earth Syst. Sci. Data, 17(5), 2135-2145. https://doi.org/10.5194/essd-17-2135-2025

• Zhang, Y., & Wu, X. (2025). Global space-time patterns of sub-daily extreme precipitation. Environmental Research Letters, 20. https://doi.org/10.1088/1748-9326/ade607

• D Silva, S., Mathew, B. P., V, K. V., Bhadran, A., Girishbai, D.,... Gopinath, G. (2024). Geomorphic evolution of a tropical river basin. Geology, Ecology, and Landscapes, 1–12. https://doi.org/10.1080/24749508.2024.2359787

• Abegeja, D., & Nedaw, D. (2024). Identification of groundwater potential zones in Meki Catchment. Geology, Ecology, and Landscapes, 1–16. https://doi.org/10.1080/24749508.2024.2392380

• Ma, Q., Qian, Y., Su, W., Shi, L., Wang, E., Yu, A.,... Lu, Y. (2025). Degradation of agricultural polyethylene film by greater wax moth larvae. Ecotoxicology and Environmental Safety, 303, 118841. https://doi.org/10.1016/j.ecoenv.2025.118841

• Sun, S., Xie, W., Wang, G., Zhang, W., Hu, Z., Sun, X.,... DeLuca, T. H. (2025). Evidence for phosphorus cycling parity in N2-fixing pioneer plant species. Functional Ecology, 39(4), 985-1000. https://doi.org/10.1111/1365-2435.70023

• Zhu, Z., Lu, R., Yu, B., Li, T., & Yeh, S. (2024). A moderator of tropical impacts on climate in Canadian Arctic Archipelago. Nature Communications, 15(1), 8644. https://doi.org/10.1038/s41467-024-53056-0

Reviewer #4: 1. Kindly highlight the significance of study in last paragraph of Introduction.

2. Provide a robust conclusion, which paves the way for future scientific studies in this direction.

3. You may discuss, latest current work of this field with emphasis on microbial population of springs.

**Do you want your identity to be public for this peer review?** For information about this choice, including consent withdrawal, please see our Privacy Policy

Reviewer #1: No

Reviewer #2: **Yes:** Patrick Omoregie Isibor

Reviewer #3: **Yes:** Pavan Kumar

Reviewer #4: **Yes:** Jabrinder Singh

---

## [Author Response · Author response to Decision Letter 1]

4 Dec 2025

Please see the attached file entitled Revisions Checklist and Reasoning for a detailed list of every reviewer comment and how the authors addressed them. Thank you to the editors and reviewers for their time and commitment to this manuscript.

---

## [Decision Letter · Decision Letter 1]

30 Jan 2026

A comparison of microbial community composition in two alpine springs in southern Nevada

PONE-D-25-05980R1

Dear Dr. Bouali,

We’re pleased to inform you that your manuscript has been judged scientifically suitable for publication and will be formally accepted for publication once it meets all outstanding technical requirements.

Kind regards,

Renjith VishnuRadhan, PhD

Academic Editor

PLOS One

Reviewers' comments:

Reviewer's Responses to Questions

**Comments to the Author**

Reviewer #3: All comments have been addressed

2. Is the manuscript technically sound, and do the data support the conclusions?

Reviewer #3: Yes

3. Has the statistical analysis been performed appropriately and rigorously?

Reviewer #3: Yes

4. Have the authors made all data underlying the findings in their manuscript fully available?

Reviewer #3: Yes

5. Is the manuscript presented in an intelligible fashion and written in standard English?

Reviewer #3: Yes

Reviewer #3: Accepted. The manuscript has been revised thoroughly, and all reviewer comments have been carefully addressed.

**Do you want your identity to be public for this peer review?** For information about this choice, including consent withdrawal, please see our Privacy Policy

Reviewer #3: **Yes:** Pavan Kumar

---

## [Editor Report · Acceptance letter]

PONE-D-25-05980R1

PLOS One

Dear Dr. Bouali,

I'm pleased to inform you that your manuscript has been deemed suitable for publication in PLOS One. Congratulations! Your manuscript is now being handed over to our production team.

Kind regards,

on behalf of

Dr. Renjith VishnuRadhan

Academic Editor

PLOS One